# Prediction of metastatic prostate cancer by prostate-specific antigen in combination with T stage and Gleason Grade: Nationwide, population-based register study

**Frederik B. Thomsen**[ID][1]*, **Marcus Westerberg**[ID][2,3], **Hans Garmo**[2,4], **David Robinson**[ID][5], **Lars Holmberg**[2], **Hans David Ulmert**[6,7,8], **Pär Stattin**[2]

**1** Copenhagen Prostate Cancer Center, Department of Urology, Rigshospitalet, University of Copenhagen, Copenhagen, Denmark, **2** Department of Surgical Sciences, Uppsala University Hospital, Uppsala, Sweden, **3** Department of Mathematics, Uppsala University, Uppsala, Sweden, **4** King's College London, School of Medicine, Division of Cancer Studies, Cancer Epidemiology Group, London, United Kingdom, **5** Department of Urology, Ryhov Hospital, Jönköping, Sweden, **6** University of California Los Angeles, Department of Molecular and Medical Pharmacology, Los Angeles, CA, United States of America, **7** Ahmanson Translational Imaging Division, David Geffen UCLA School of Medicine, Los Angeles, CA, United States of America, **8** Jonsson Comprehensive Cancer Center, David Geffen UCLA School of Medicine, Los Angeles, CA, United States of America

* thomsen.frederik@gmail.com

**Data Availability Statement:** The Research Ethics Board at Uppsala University approved of the linkages in our project (PCBaSe 2016-239). We

## Abstract

The objective was to investigate the proportion of men with metastatic prostate cancer in groups defined by T stage, Gleason Grade Group (GGG) and serum levels of prostate-specific antigen (PSA) and if PSA can be used to rule in metastatic prostate cancer when combined with T stage and GGG. We identified 102,076 men in Prostate Cancer data Base Sweden 4.0 who were diagnosed with prostate cancer in 2006–2016. Risk of metastases was assessed for PSA stratified on T stage and five-tiered GGG. For men who had not undergone bone imaging, we used multiple imputation to classify metastatic prostate cancer. Advanced T stage, high GGG and high PSA were related to bone metastases. For example: only 79/38 190 (0.2%) of men with T1-2 and GGG 1 had PSA above 500 ng/mL, and 29/79 (44%) of these men had metastases; whereas 1 154/7 018 (16%) of men with T3-4 and GGG 5 had PSA above 500 ng/ml and 1 088/1 154 (94%) of these men had metastases. However, no PSA cut-off could accurately identify the majority of men with metastatic prostate cancer (i.e. high sensitivity) while also correctly classifying most men without metastasis (i.e. high specificity). In conclusion, these results support the use of imaging to confirm bone metastases in men with advanced prostate cancer as no PSA level in combination with T stage and GGG could accurately rule in metastatic prostate cancer and thereby safely omit bone imaging.

## Introduction

Advanced T stage, high Gleason grades and high serum levels of prostate-specific antigen (PSA) are associated with bone metastases in men with prostate cancer [1,2]. Based on results

received a study file from Statistics Sweden and the National Board of Health and Welfare where the person identity number for men in the National prostate Cancer Register had been replaced by a code. This means that the data set is pseudononymized, but due to the large number of variables this dataset is still considered not anonymized when deleting this code. The following restrictions apply: we are not allowed to share data on individuals with other researchers, nor or we allowed upload such data on an open server. However, we can provide access to the dataset on a remote server on demand. On the Research platform, data can be uploaded and then accessed by external researchers. However, no individual data are allowed to leave the platform but aggregated data in the form of figures and Tables can be exported. This research project has been approved by by the Research Ethics Board in Uppsala (dnr 2016- 239) with Pär Stattin as contact person. External researchers who wish to access data should contact the Research Ethics Board mail: registrator@etikprovning.se | Postal adress: Etikprövningsnämnden, Box 2110, Uppsala, Sweden.

**Funding:** This project received research support from the Swedish Cancer Society (2016-464) and (16 0700) to PS and the Swedish Research Council (2017-00847) to PS. FBT is supported by research grant from IMK Almene Fond. The funders had no role in study design, data collection and analysis, decision to publish, or preparation of the manuscript.

**Competing interests:** The authors have declared that no competing interests exist.

from some small, single centre studies published in the beginning of the 1990s, PSA levels above 100 ng/mL have been used as a proxy for metastatic prostate cancer [3–5]. However, identification and quantification of metastases is increasingly important since radical treatment in men with very high-risk, non-metastatic prostate cancer have been suggested to be beneficial in observational studies [6–8] and quantification of tumour extent is used as a basis for selection of novel treatments in addition to androgen deprivation therapy in men with advanced prostate cancer [9–11].

Recently, two observational studies reported that among men with PSA 100 ng/mL or higher who had undergone bone imaging, only 45–75% had metastatic prostate cancer [12,13]. These studies were hampered by few men with PSA >100 ng/mL (n = 241) [13] or the possibility of selection bias since only men who had undergone imaging were included in a previous PCBaSe study (7 521 out of 15 635 men with PSA above 100 ng/mL) [12]. To avoid this shortcoming we applied imputation of metastatic status in men who had not undergone bone imaging and in addition both T stage and Gleason grades should be included in the model since these factors are also predictive of metastases. The aim of this study was to investigate the proportion of men with metastatic prostate cancer in groups defined by T stage, Gleason Grade Group (GGG) and serum levels of prostate-specific antigen (PSA) and if PSA can be used to rule in metastatic prostate cancer when combined with T stage and GGG.

## Material and methods

Prostate Cancer data Base Sweden (PCBaSe) 4.0 contains information on cancer characteristics at diagnosis and primary treatment from the National Prostate Cancer Register (NPCR) of Sweden. Since 1998, NPCR captures 98% of all incident prostate cancer cases in the National Cancer Registry to which recording is mandated by law [14]. In addition, data from a number of other health care registers and demographic databases have been obtained by linkage using the individual unique Swedish person identity number. Comorbidity was assessed by use of Charlson comorbidity index (CCI) based on discharge diagnoses in The Patient Registry as previously described [14]. Educational level, income, and marital status were assessed by use of data in the LISA database, a socioeconomic database, and cause and date of death were obtained from The Cause of Death Registry [15–17]. This study was approved by the Research Ethics Board at Uppsala University and included all men diagnosed with prostate cancer in 2006–2016 in NPCR.

The following variables in PCBaSe were used: age and year of prostate cancer diagnosis, mode of detection, clinical TNM, serum PSA level (ng/ml), Gleason grading of the diagnostic biopsy reported with the five-tiered Gleason Grade Groups (GGG) [18], primary treatment, CCI, educational level, and marital status.

In NPCR, M stage has been recorded as M1, M0, or Mx with M1 indicating bone metastases on imaging, M0 indicating no bone metastases on imaging and Mx indicating no imaging performed. Current Swedish guidelines recommends against bone imaging in men with low or intermediate-risk ('favourable-risk') prostate cancer, i.e. T1-2, PSA less than 20 ng/mL and GGG 1–2 [19]. Imaging was primarily performed with a $^{99m}$Technetium-radiolabeled bisphosphonate bone scan. In 2009 this imaging modality compromised approximately 95% of all bone imaging in Sweden, which in 2017 was 80%.

### Statistics

Multiple imputation was implemented using the method of Chained Equations (MICE) [20]. A series of univariate marginal models were specified to impute each of the variables with missing data: TNM stage, PSA, GGG, mode of detection, primary treatment, civil status, and

education level. The imputation model also included age at diagnosis, year of diagnosis, cause of death/censoring and follow up time for which there was complete capture in the database.

We aimed to impute the clinical stages 'metastatic prostate cancer' and 'non-metastatic prostate cancer', corresponding to the results of a bone imaging under the hypothetical scenario in which all men had undergone such an investigation. The imputation procedure has previously been applied and described and the number of multiple imputations was set to 20 with 10 iterations [21].

All following analyses were conducted based on the imputed datasets by combining estimates from each completed dataset using Rubin's rules. Men with stage T1a-b, i.e. diagnosed at transurethral resection of the prostate, were excluded from further analyses. The proportion of men with metastatic prostate cancer in different PSA ranges (<20, 20-<50, 50-<100, 100-<300, 300-<500 and >500 ng/ml) was calculated stratified on T stage and GGG. To assess different PSA cut-offs (20, 50, 100, 200, 300, 500, 1000) to predict metastatic prostate cancer we calculated sensitivity and specificity as well as ROC curves, positive and negative predictive value and likelihood ratios. Likelihood ratios are calculated using sensitivity and specificity to assess if a test usefully changes the probability of a condition–in this case presence of metastases. A positive likelihood ratio is defined as the true positive rate divided by false positive rate, whereas the negative likelihood ratio is the false positive rate divided by the true negative rate. Likelihood ratios range from 0 to infinity with a result of 1 indicating no diagnostic value. A high positive likelihood ratio suggests that metastases are present, while a low negative likelihood test suggests there are few false negative cases compared to true negative cases [22]. Fagan's nomograms based on positive likelihood ratios for cut-offs 100 and 500 ng/mL were created. The analyses were performed using R 3.4.2.

## Results and discussion

Baseline characteristics of 102 077 men included in the study are presented in Table 1. In total 30 426 men (30%) had undergone imaging at diagnosis. The majority, 45 965 / 70 313 (66%) of men who had not undergone imaging had favourable-risk prostate cancer–i.e. T1-2, PSA less than 20 ng/ml and GGG 1–2 –and had thus been perceived to have non-metastatic prostate cancer and therefor imaging had not been performed in accordance with Swedish guidelines. After imputation, an average of 4 870 (7%) of men who had not undergone imaging were classified with metastatic prostate cancer.

Advanced T stage, higher GGG and higher PSA were strongly related to the presence of bone metastases. Fig 1 depicts the proportion of men with metastatic prostate cancer depending on GGG and PSA. The proportion of men with metastatic prostate cancer increased with advanced T stage higher GGG and higher PSA, S1 Table. For example, in men with T1-2, PSA 100–300 ng/mL and GGG 1, 10% had metastatic prostate cancer, whereas in men with T3-4, PSA 100–300 ng/mL and GGG 5, 71% had metastases. In men with T1-2, PSA below 20 ng/mL and GGG 3, 3% had bone metastases, whereas in men with T3-4, PSA above 500 ng/mL and GGG 3, 91% had metastases. Of all men with PSA above 100, only 71% (6246 / 8850) had metastatic prostate cancer.

The Fagan's nomogram for positive likelihood ratio of the probability of metastases for men with PSA between 100 ng/mL and 500 ng/mL stratified on T stage and GGG is depicted in Fig 2. The pre-test probability of metastatic disease corresponds to the prevalence defined by T stage and GGG, not considering PSA at diagnosis, whilst the post-test probability also incorporates PSA. For example, in men with T1-2 and GGG 5 the proportion of metastases was 30%, while the post-test probability was 70% for PSA 100 ng/mL as cut-off and 92% for PSA 500 ng/mL. For men with T3-4 and GGG 5 the corresponding proportion was 58% and

**Table 1. Baseline characteristics of 106 932 men diagnosed with prostate cancer in 2006–2016 in PCBaSe 4.0.**

| | M0 | | M1 | | Mx | | All | |
|---|---|---|---|---|---|---|---|---|
| | n | (%) | n | (%) | n | (%) | n | (%) |
| **Total** | **21885** | **(100)** | **8541** | **(100)** | **71651** | **(100)** | **102077** | **(100)** |
| Age at diagnosis, years | | | | | | | | |
| 60 or less | 3046 | (13) | 619 | (8) | 12657 | (18) | 16322 | (16) |
| 61–70 | 10170 | (42) | 2269 | (30) | 30557 | (43) | 42996 | (42) |
| 71–80 | 8655 | (36) | 2776 | (36) | 18781 | (27) | 30212 | (30) |
| 81 or older | 2261 | (9) | 1967 | (26) | 8318 | (12) | 12546 | (12) |
| Year of diagnosis | | | | | | | | |
| 2006 | 1967 | (8) | 671 | (9) | 6015 | (9) | 8653 | (8) |
| 2007 | 1613 | (7) | 625 | (8) | 6285 | (9) | 8523 | (8) |
| 2008 | 1387 | (6) | 623 | (8) | 6362 | (9) | 8372 | (8) |
| 2009 | 1830 | (8) | 682 | (9) | 7528 | (11) | 10040 | (10) |
| 2010 | 2082 | (9) | 654 | (9) | 6559 | (9) | 9295 | (9) |
| 2011 | 855 | (4) | 255 | (3) | 7976 | (11) | 9086 | (9) |
| 2012 | 2108 | (9) | 670 | (9) | 5782 | (8) | 8560 | (8) |
| 2013 | 2433 | (10) | 807 | (11) | 5928 | (8) | 9168 | (9) |
| 2014 | 2869 | (12) | 858 | (11) | 6732 | (10) | 10459 | (10) |
| 2015 | 3250 | (13) | 883 | (12) | 5827 | (8) | 9960 | (10) |
| 2016–2017 | 3738 | (15) | 903 | (12) | 5319 | (8) | 9960 | (10) |
| Educational level | | | | | | | | |
| Low | 8377 | (35) | 3308 | (43) | 23189 | (33) | 34874 | (34) |
| Intermediate | 9570 | (40) | 2795 | (37) | 27920 | (40) | 40285 | (39) |
| High | 6023 | (25) | 1445 | (19) | 18649 | (27) | 26117 | (26) |
| Missing | 162 | (1) | 83 | (1) | 555 | (1) | 800 | (1) |
| Civil status | | | | | | | | |
| Widow last | 1786 | (7) | 907 | (12) | 5331 | (8) | 8024 | (8) |
| Married/partnership | 15748 | (65) | 4621 | (61) | 46619 | (66) | 66988 | (66) |
| Unmarried | 2852 | (12) | 1008 | (13) | 7854 | (11) | 11714 | (11) |
| Divorced/separated | 3735 | (15) | 1092 | (14) | 10467 | (15) | 15294 | (15) |
| Missing | 11 | (0) | 3 | (0) | 42 | (0) | 56 | (0) |
| Charlson Comorbdiity Index (CCI) | | | | | | | | |
| CCI 0 | 18337 | (76) | 4885 | (64) | 54637 | (78) | 77859 | (76) |
| CCI 1 | 3342 | (14) | 1313 | (17) | 8392 | (12) | 13047 | (13) |
| CCI 2 | 1433 | (6) | 664 | (9) | 4101 | (6) | 6198 | (6) |
| CCI 3+ | 1020 | (4) | 769 | (10) | 3183 | (5) | 4972 | (5) |
| Symptoms preceding work-up | | | | | | | | |
| Asymptomatic | 10927 | (45) | 1247 | (16) | 34202 | (49) | 46376 | (45) |
| Symptomatic | 8496 | (35) | 2676 | (35) | 21998 | (31) | 33170 | (32) |
| Missing | 4709 | (20) | 3708 | (49) | 14113 | (20) | 22530 | (22) |
| T stage | | | | | | | | |
| T1 | 8012 | (33) | 563 | (7) | 40119 | (57) | 48694 | (48) |
| T2 | 9462 | (39) | 1622 | (21) | 20051 | (29) | 31135 | (31) |
| T3 | 5828 | (24) | 3702 | (49) | 7097 | (10) | 16627 | (16) |
| T4 | 422 | (2) | 1446 | (19) | 1200 | (2) | 3068 | (3) |
| Missing | 408 | (2) | 298 | (4) | 1846 | (3) | 2552 | (3) |
| N stage | | | | | | | | |
| N0 | 10328 | (43) | 958 | (13) | 8016 | (11) | 19302 | (19) |

(*Continued*)

**Table 1.** (Continued)

| | M0 | | M1 | | Mx | | All | |
|---|---|---|---|---|---|---|---|---|
| | **n** | **(%)** | **n** | **(%)** | **n** | **(%)** | **n** | **(%)** |
| **Total** | **21885** | **(100)** | **8541** | **(100)** | **71651** | **(100)** | **102077** | **(100)** |
| N1 | 1425 | (6) | 1181 | (15) | 523 | (1) | 3129 | (3) |
| Nx | 12364 | (51) | 5484 | (72) | 61537 | (88) | 79385 | (78) |
| Missing | 15 | (0) | 8 | (0) | 237 | (0) | 260 | (0) |
| Gleason Grade Group (GGG) | | | | | | | | |
| GGG 1 | 3721 | (15) | 199 | (3) | 35158 | (50) | 39078 | (38) |
| GGG 2 | 6082 | (25) | 500 | (7) | 17131 | (24) | 23713 | (23) |
| GGG 3 | 5330 | (22) | 1066 | (14) | 7114 | (10) | 13510 | (13) |
| GGG 4 | 4395 | (18) | 1825 | (24) | 4499 | (6) | 10719 | (11) |
| GGG 5 | 4104 | (17) | 3154 | (41) | 3868 | (6) | 11126 | (11) |
| Missing | 500 | (2) | 887 | (12) | 2543 | (4) | 3930 | (4) |
| PSA, ng/ml | | | | | | | | |
| Median (Q1-Q3) | 14 | (8–30) | 133 | (39–503) | 8 | (5–13) | 9 | (6–22) |
| 0–19 | 14711 | (61) | 1067 | (14) | 57496 | (82) | 73274 | (72) |
| 20–49 | 5850 | (24) | 1128 | (15) | 6109 | (9) | 13087 | (13) |
| 50–99 | 2016 | (8) | 1075 | (14) | 2480 | (4) | 5571 | (5) |
| 100–199 | 1100 | (5) | 1674 | (22) | 1679 | (2) | 4453 | (4) |
| 200–399 | 158 | (1) | 688 | (9) | 426 | (1) | 1272 | (1) |
| 400+ | 138 | (1) | 1930 | (25) | 903 | (1) | 2971 | (3) |
| Missing | 159 | (1) | 69 | (1) | 1220 | (2) | 1448 | (1) |
| Treatment | | | | | | | | |
| Androgen deprivation therapy | 6454 | (27) | 7178 | (94) | 13296 | (19) | 26928 | (26) |
| Watchfull waiting | 631 | (3) | 81 | (1) | 4572 | (7) | 5284 | (5) |
| Active surveillance | 1260 | (5) | 41 | (1) | 18204 | (26) | 19505 | (19) |
| Radical prostatectomy | 6273 | (26) | 48 | (1) | 20659 | (29) | 26980 | (26) |
| Radiotherapy | 8496 | (35) | 117 | (2) | 6987 | (10) | 15600 | (15) |
| Chryotherapy | 309 | (1) | 19 | (0) | 2607 | (4) | 2935 | (3) |
| Missing | 709 | (3) | 147 | (2) | 3988 | (6) | 4844 | (5) |

the post-test probabilities for PSA 100 and 500 were 79% and 94%, respectively. The positive likelihood ratios in the Fagan nomograms were derived from ROC curves for predicting metastases (S1 Fig). The overall AUC varied between 0.76 and 0.88, depending on T stage and GGG. In general, the positive likelihood ratio decreased with higher GGG and increased with higher PSA cutoff, as shown in S2 and S3 Tables, where eight PSA cut-offs were investigated. In no subgroup could any PSA cut-off identify the majority of men with metastatic prostate cancer (i.e. high sensitivity) while also correctly classifying most men without metastasis (i.e. high specificity).

## Discussion

In this nationwide, population-based cohort study, the proportion of men with metastatic prostate cancer increased with higher T stage, Gleason grade and PSA. However, no PSA level in combination with T stage and GGG could accurately rule in metastatic prostate cancer and thereby safely omit bone imaging. Arguably, the best proxy for ruling in metastatic prostate cancer was a PSA level above 500 ng/mL, however this group merely comprised 3% of the

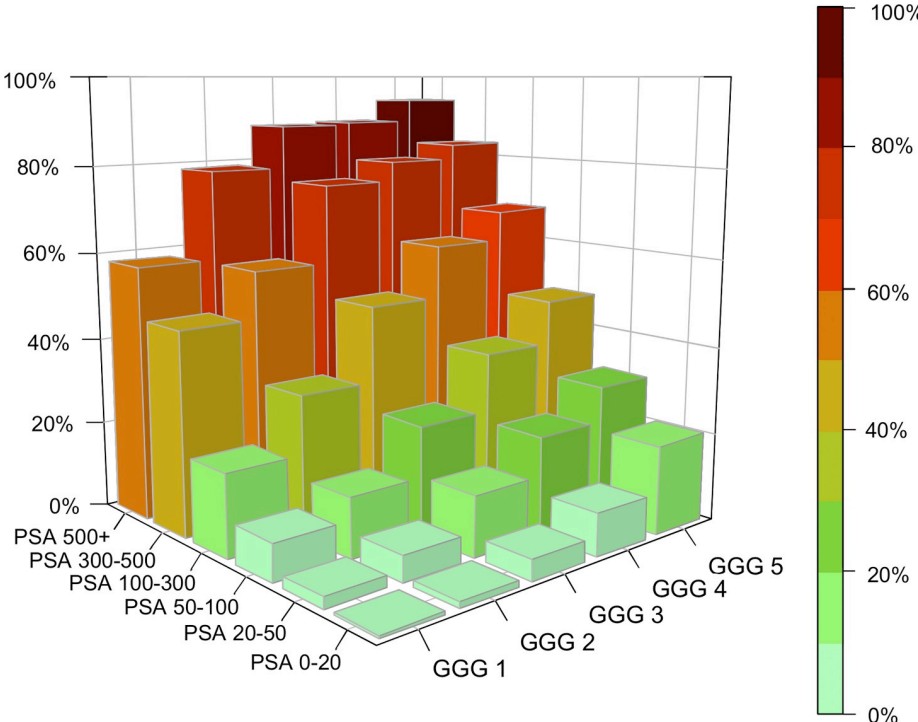

**Fig 1. Proportion and number of men with metastases stratified by prostate-specific antigen (PSA) and Gleason Grade Groups (GGG).**

study population. Even for this PSA cut-off the post-test probability of metastatic prostate cancer ranged from 76–94% in men with T3-4 and 50–63% in men with T1-2 cancer.

In our previous study, we assessed only men who had undergone bone imaging and this may have introduced bias. To overcome this limitation, we used imputation of metastatic status in men who had not undergone diagnostic bone imaging in the current study and we also included T stage and Gleason. There is no established cut-off in the literature for an acceptable percentage of missing data in a data set for valid statistical inferences [23] and the success of an imputation depends on the knowledge on conditions under which missing data occurred and the correlation between missing data and non-missing covariates. Some factors influencing use of bone imaging are available in PCBaSe, e.g. age, comorbidity, clinical characteristics, and primary treatment whereas bone-related symptoms are incompletely reported. Furthermore, as metastatic status affects treatment selection and prostate cancer death, we argue that the statistical inference based on our data is valid despite the imputation of metastatic status in 70% of all men. Strength of our study include the comprehensive data from high quality health care registers and demographic databases [14–16].

The use of PSA above 100 ng/mL as a proxy for metastatic prostate cancer is based on some small, single-centre studies performed in the 1990s including less than 200 men with prostate cancer [3–5]. In these studies, a PSA threshold of 100 ng/mL had a positive predictive value for metastasis of 94–100%. In two more recent studies the predictive value was considerably lower. In the previous PCBaSe study including men diagnosed in 1998–2009, 25% of men who underwent diagnostic imaging with PSA above 100 ng/mL did not have metastases on imaging [12]. Similarly, in an Australian single centre study 45% (109 / 241) of men diagnosed in 1998–2013 with PSA above 100 ng/ml had no metastases on bone imaging [13]. In our present study, the positive predictive value was 22–76% and was strongly dependent on T stage and

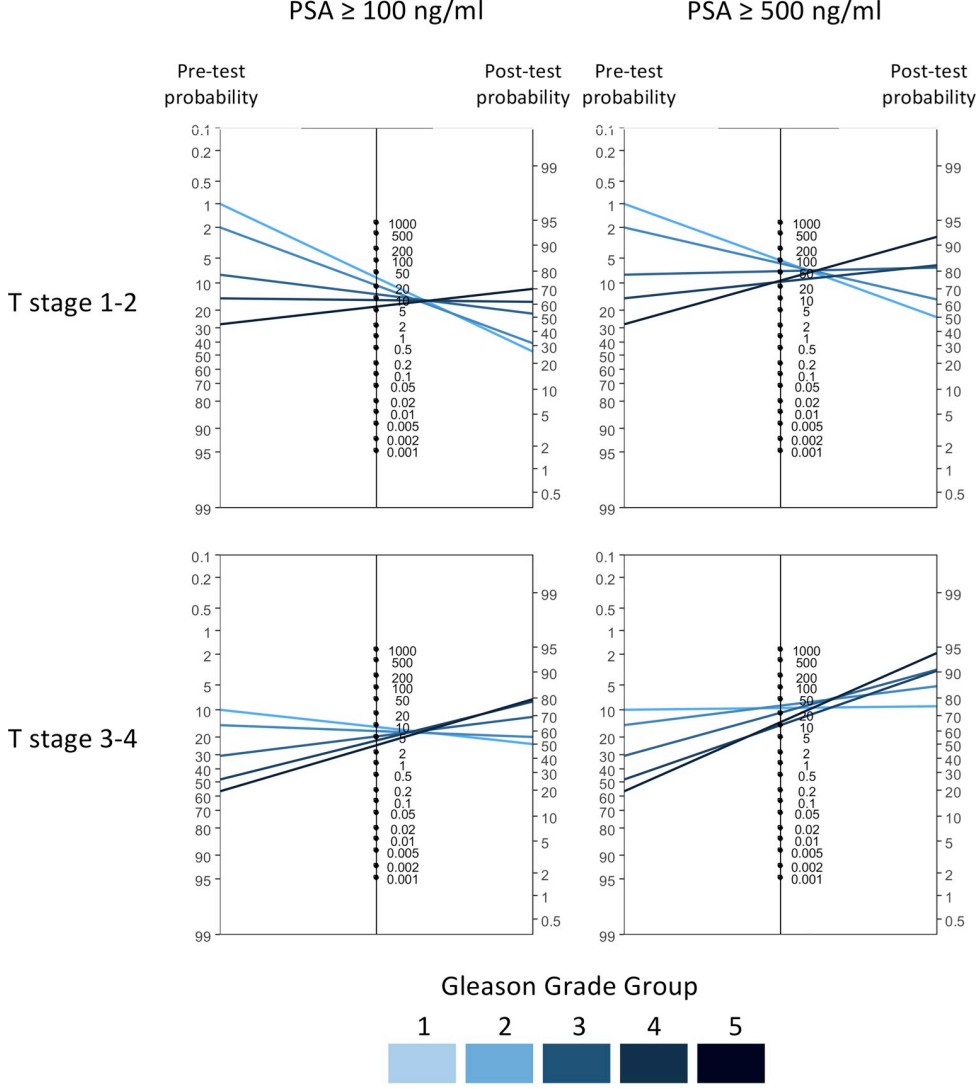

**Fig 2. Fagan's nomogram for calculation of post-test probabilities of metastatic prostate cancer.** Lines indicate post-test probability of metastatic prostate cancer for men with prostate-specific antigen (PSA) above given cut-off related to the corresponding pre-test probability (prevalence) by T stage and Gleason Grade Group.

Gleason grade. Moreover, the specificity for PSA 100 ng/mL was low, so a large number of men without metastatic disease would be misclassified if this PSA level was used to define presence of bone metastasis. The most likely explanation for the difference in predictive value between the studies from the nineties and the more current studies is a stage migration caused by increased use of PSA testing [21,24].

The Swedish guidelines for prostate cancer care recommend against bone imaging in men with a low risk of metastatic prostate cancer and the adherence to this recommendation has become high [19]. Imaging in men with low-risk prostate cancer decreased from 45% in 1998 to 3% in 2009, while for intermediate-risk prostate cancer the corresponding decrease was from 58% to 16%. In the latter category, men with GGG 3 are more likely to undergo imaging compared to men with GGG 2 [25]. As a consequence of this selection, the majority of men (66%) with unknown metastatic status had a favourable-risk prostate cancer and in these men

the prevalence of metastatic prostate cancer was very low (<1%) supporting the Swedish guidelines.

A recent study from PCBaSe reported that age was not a strong independent risk factor for prostate cancer death, contrary to previous assumptions [24] but old men were less likely to undergo adequate diagnostic workup including bone imaging. Radical treatment may possibly be beneficial in men with PSA higher than 100 ng/mL as suggested by two recent observational studies in PCBaSe [8,26], however, there are no data from RCT in support of these observations. Correct staging is also important in men in whom hormonal therapy is considered since men with non-metastatic, locally advanced prostate cancer can be safely managed with antiandrogen monotherapy [27,28], whereas men with metastases are best managed with castration therapy [29]. Finally, tumour extent on bone imaging has recently become an indication for additional treatment to androgen deprivation therapy in men with metastatic prostate cancer [9–11].

## Conclusion

In this nationwide population-based study, metastatic prostate cancer could not be ruled in with sufficient accuracy by any combination of T stage, Gleason grade and PSA. The best PSA cut-off for predicting metastases was 500 ng/mL. However, even for this very high cut-off risk of metastases ranged from 50–94% dependent on T stage and Gleason grade. Our results emphasize the importance of bone imaging in men with advanced prostate cancer for correct staging as a basis for optimal treatment selection.

## Supporting information

**S1 Fig. ROC curves for predicting metastases for all men and stratified by T stage and GGG (Gleason Grade Groups).**
(PDF)

**S1 Table. Total number of men in subgroups defined by prostate-specific antigen (PSA), Gleason Grade Groups (GGG) and T stage and number of men with metastatic prostate cancer (percentage of men with metastases with metastatic prostate cancer in the subgroup).**
(DOCX)

**S2 Table. Sensitivity, 1-specificity, positive predictive value (PPV), negative predictive value (NPV), Positive likelihood ratio (LRH+), negative likelihood ratio (LRH-) for predicting metastases in men with T1-2 prostate cancer.**
(DOCX)

**S3 Table. Sensitivity, 1-specificity, positive predictive value (PPV), negative predictive value (NPV), Positive likelihood ratio (LRH+), negative likelihood ratio (LRH-) for predicting metastases in men with T3-4 prostate cancer.**
(DOCX)

## Acknowledgments

This project received research support from the Swedish Cancer Society (2016–464) and the Swedish Research Council (2017–00847). This project was made possible by the continuous work of the National Prostate Cancer Register of Sweden (NPCR) steering group: Pär Stattin (chair), Ingela Franck Lissbrant (deputy chair), Johan Styrke, Camilla Thellenberg Karlsson, Lennart Åström, Stefan Carlsson, Marie Hjälm-Eriksson, David Robinson, Mats Andén, Ola

Bratt, Magnus Törnblom, Johan Stranne, Jonas Hugosson, Maria Nyberg, Olof Akre, Per Fransson, Eva Johansson, Calle Waller, Gert Malmberg, Fredrik Sandin, and Karin Hellström.

## Author Contributions

**Conceptualization:** Frederik B. Thomsen, Hans Garmo, Pär Stattin.

**Data curation:** David Robinson, Pär Stattin.

**Formal analysis:** Marcus Westerberg, Hans Garmo.

**Methodology:** Hans Garmo, Pär Stattin.

**Resources:** Pär Stattin.

**Writing – original draft:** Frederik B. Thomsen.

**Writing – review & editing:** Marcus Westerberg, Hans Garmo, David Robinson, Lars Holmberg, Hans David Ulmert, Pär Stattin.

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
