## [Decision Letter · Decision Letter 0]

10 Dec 2019

PONE-D-19-30700

Prediction of metastatic prostate cancer by prostate-specific antigen in combination with T stage and Gleason Grade. Nationwide, population-based register study

PLOS ONE

Dear Dr. Thomsen,

Thank you for submitting your manuscript to PLOS ONE. After careful consideration, we feel that it has merit but does not fully meet PLOS ONE’s publication criteria as it currently stands. Therefore, we invite you to submit a revised version of the manuscript that addresses the points raised during the review process.

The image quality in Figure 1 needs to be improved.

We would appreciate receiving your revised manuscript by Jan 24 2020 11:59PM. To enhance the reproducibility of your results, we recommend that if applicable you deposit your laboratory protocols in protocols.io, where a protocol can be assigned its own identifier (DOI) such that it can be cited independently in the future. For instructions see: http://journals.plos.org/plosone/s/submission-guidelines#loc-laboratory-protocols

We look forward to receiving your revised manuscript.

Kind regards,

Lucia R. Languino, Ph.D.

Academic Editor

PLOS ONE

2. We noticed you have some minor occurrence(s) of overlapping text with the following previous publication(s), which needs to be addressed:

http://dx.doi.org/10.1016/j.eururo.2017.06.036

https://doi.org/10.1080/0284186X.2019.1662084

https://doi.org/10.1111/bju.14563

In your revision ensure you cite all your sources (including your own works), and quote or rephrase any duplicated text outside the Methods section. Further consideration is dependent on these concerns being addressed.

3. In the ethics statement in the manuscript and in the online submission form, please provide additional information about the patient records/samples used in your retrospective study. Specifically, please ensure that you have discussed whether all data/samples were fully anonymized before you accessed them and/or whether the IRB or ethics committee waived the requirement for informed consent. If patients provided informed written consent to have data/samples from their medical records used in research, please include this information.

Reviewers' comments:

Reviewer's Responses to Questions

**Comments to the Author**

1. Is the manuscript technically sound, and do the data support the conclusions?

Reviewer #1: Yes

Reviewer #2: Yes

2. Has the statistical analysis been performed appropriately and rigorously? 

Reviewer #1: Yes

Reviewer #2: Yes

3. Have the authors made all data underlying the findings in their manuscript fully available?

Reviewer #1: Yes

Reviewer #2: Yes

4. Is the manuscript presented in an intelligible fashion and written in standard English?

Reviewer #1: Yes

Reviewer #2: Yes

5. Review Comments to the Author

Reviewer #1: “Prediction of metastatic prostate cancer by prostate-specific antigen in combination with T stage and Gleason Grade. Nationwide, population-based register study “

Thomsen et al.

Submitted to PLOS One

Higher T stage, Gleason grade, and serum PSA have previously been associated with bone metastases in prostate cancer patients. PSA above 100ng/mL has been used as a surrogate for metastatic prostate cancer, based on previous studies. However, two recent imaging studies suggested that PSA above 100ng/mL was associated with only a 45-75% presence of metastasis, and there are some concerns with selection bias in these studies.

Thomsen and colleagues sought to avoid the shortcoming of previous studies, and investigated the proportion of men with metastatic prostate cancer in groups defined by T stage, Gleason Grade Group (GGG), and serum PSA levels; and to determine if PSA can rule in metastatic disease when combined with T stage and GGG.

Conclusions are: 1) Advanced T stage, higher GGG, and higher PSA were strongly related to bone metastatic prostate cancer; and 2) PSA, regardless of cut-off, could not correctly identify men with metastatic disease with either sensitivity or specificity. Based on the second conclusion, omitting bone imaging could not safely be suggested. Main take home is metastatic prostate cancer could not be ruled in with sufficient accuracy by any combination of T stage, Gleason grade and PSA.

Critique:

Authors present the data and draw conclusions nicely. The only (minor) concern is the image quality of Figure 1. If a less “fuzzy” image could be supplied, that would be beneficial.

Reviewer #2: The goal of the manuscript “Prediction of metastatic prostate cancer by prostate-specific antigen in combination with T stage and Gleason Grade. Nationwide, population-based register study” is to assess whether serum PSA levels combined with T stage and Gleason Grade can be utilized to rule in metastatic prostate cancer. The study utilized a large Swedish cohort of patients (102,076 men) diagnosed with prostate cancer between 2006 and 2016. The study is well presented and suggests a necessity of imaging to detect bone metastases in men with advanced prostate cancer as serum PSA levels in combination with T stage and Gleason score could not accurately rule in metastatic prostate cancer. Publishing such a study will be valuable.

Minor concern that needs to be addressed before publication:

1. The authors need to provide a high resolution image for Fig 1.

6. PLOS authors have the option to publish the peer review history of their article (what does this mean?). If published, this will include your full peer review and any attached files.

Reviewer #1: No

Reviewer #2: No

---

## [Author Response · Author response to Decision Letter 0]

9 Jan 2020

Dec 17 2019

Re: PONE-D-19-30700

Prediction of metastatic prostate cancer by prostate-specific antigen in combination with T stage and Gleason Grade. Nationwide, population-based register study

PLOS ONE

The image quality in Figure 1 needs to be improved.

Answer: Figure 1 with higher resolution is included in the resubmission. 

Answer: the ms has been updated accordingly

2. We noticed you have some minor occurrence(s) of overlapping text with the following previous publication(s), which needs to be addressed:

http://dx.doi.org/10.1016/j.eururo.2017.06.036

https://doi.org/10.1080/0284186X.2019.1662084

https://doi.org/10.1111/bju.14563

In your revision ensure you cite all your sources (including your own works), and quote or rephrase any duplicated text outside the Methods section. Further consideration is dependent on these concerns being addressed.

Answer: All studies use data from PCBaSe and there are therefore some parts of the methods section that is quite similar but where we reference other cohort profile publications. We reread all three papers and compared to the current study. We have not been able to find any overlapping text outside the methods section. If there are any sections that you deem necessary to reference please inform us of any overlapping text. Thank you. 

3. In the ethics statement in the manuscript and in the online submission form, please provide additional information about the patient records/samples used in your retrospective study. Specifically, please ensure that you have discussed whether all data/samples were fully anonymized before you accessed them and/or whether the IRB or ethics committee waived the requirement for informed consent. If patients provided informed written consent to have data/samples from their medical records used in research, please include this information.

Answer:

Our statement regarding ethics in our submitted ms read: “This study was approved by the Research Ethics Board at Uppsala University and included all men diagnosed with prostate cancer in 2006-2016 in NPCR.

The Research Ethics Board at Uppsala University approved of the linkages in our project (PCBaSe 2016-239). We received a study file from Statistics Sweden and the National Board of Health and Welfare where the person identity number for men in the National prostate Cancer Register had been replaced by a code. This means that the data set is pseudononymized, but due to the large number of variables this dataset is still considered not anonymized when deleting this code. The following restrictions apply: we are not allowed to share data on individuals with other researchers, nor or we allowed upload such data on an open server. However, we can provide access to the dataset on a remote server on demand. On the Research platform, data can be uploaded and then accessed by external researchers. However, no individual data are allowed to leave the platform but aggregated data in the form of figures and Tables can be exported. 

External researcher should contact the corresponding author who will direct the demand for data to the PCBaSe reference group who will then provider the access described above. 

RE: See above 

RE: See above

Answer: the ms has been updated accordingly

---

## [Editor Report · Decision Letter 1]

16 Jan 2020

Prediction of metastatic prostate cancer by prostate-specific antigen in combination with T stage and Gleason Grade. Nationwide, population-based register study

PONE-D-19-30700R1

Dear Dr. Thomsen,

We are pleased to inform you that your manuscript has been judged scientifically suitable for publication and will be formally accepted for publication once it complies with all outstanding technical requirements.

With kind regards,

Lucia R. Languino, Ph.D.

Academic Editor

PLOS ONE
---

## [Editor Report · Acceptance letter]

21 Jan 2020

PONE-D-19-30700R1 

Prediction of metastatic prostate cancer by prostate-specific antigen in combination with T stage and Gleason Grade. Nationwide, population-based register study 

Dear Dr. Thomsen:

I am pleased to inform you that your manuscript has been deemed suitable for publication in PLOS ONE. Congratulations! Your manuscript is now with our production department. 

With kind regards,

on behalf of

Dr. Lucia R. Languino 

Academic Editor

PLOS ONE